# Remodeling Lipids in the Transition from Chronic Liver Disease to Hepatocellular Carcinoma

**DOI:** 10.3390/cancers13010088

**Published:** 2020-12-30

**Authors:** Israa T. Ismail, Ashraf Elfert, Marwa Helal, Ibrahim Salama, Hala El-Said, Oliver Fiehn

**Affiliations:** 1National Liver Institute, Menoufia University, Shebin El-Kom 55955, Egypt; imismail@ucdavis.edu (I.T.I.); ashraf.yousif@liver.menofia.edu.eg (A.E.); marwa.helal@liver.menofia.edu.eg (M.H.); ibrahiem.a.alqader11@liver.menofia.edu.eg (I.S.); hala.elsaid@liver.menofia.edu.eg (H.E.-S.); 2NIH West Coast Metabolomics Center, University of California Davis, Davis, CA 95616, USA

**Keywords:** cirrhosis, polyunsaturated fatty acids, palmitate, desaturase, lipoproteins

## Abstract

**Simple Summary:**

Hepatocellular carcinoma (HCC) has poor prognosis. We studied blood lipids by comparing healthy volunteers to patients with chronic liver disease (CLD), and to patients with HCC caused by viral infections. We contrasted our findings in blood to lipid alterations in liver tumor and nontumor tissue samples from HCC patients. In blood, most lipid species were found at increased levels in CLD patients compared to healthy volunteers. This trend was mostly reversed in HCC versus CLD patients. In liver tumor tissues, levels of many lipids were decreased compared to paired nontumor liver tissues. Differences in lipid levels were further defined by alterations in the degree of saturation in the fatty acyl chains. Some lipids, including free fatty acids, saturated lysophosphatidylcholines and saturated triacylglycerides, showed a continuous trend in the transition from the blood of healthy controls to CLD and HCC patients. For HCC patients, phosphatidylglycerides showed similar alterations in both blood and tissues.

**Abstract:**

Hepatocellular carcinoma (HCC) is a worldwide health problem. HCC patients show a 50% mortality within two years of diagnosis. To better understand the molecular pathogenesis at the level of lipid metabolism, untargeted UPLC MS—QTOF lipidomics data were acquired from resected human HCC tissues and their paired nontumor hepatic tissues (*n* = 46). Blood samples of the same HCC subjects (*n* = 23) were compared to chronic liver disease (CLD) (*n* = 15) and healthy control (*n* = 15) blood samples. The participants were recruited from the National Liver Institute in Egypt. The lipidomics data yielded 604 identified lipids that were divided into six super classes. Five-hundred and twenty-four blood lipids were found as significantly differentiated (*p* < 0.05 and qFDR *p* < 0.1) between the three study groups. In the blood of CLD patients compared to healthy control subjects, almost all lipid classes were significantly upregulated. In CLD patients, triacylglycerides were found as the most significantly upregulated lipid class at qFDR *p* = 1.3 × 10^−56^, followed by phosphatidylcholines at qFDR *p* = 3.3 × 10^−51^ and plasmalogens at qFDR *p* = 1.8 × 10^-46^. In contrast, almost all blood lipids were significantly downregulated in HCC patients compared to CLD patients, and in HCC tissues compared to nontumor hepatic tissues. Ceramides were found as the most significant lipid class (qFDR *p* = 1 × 10^−14^) followed by phosphatidylglycerols (qFDR *p* = 3 × 10^−9^), phosphatidylcholines and plasmalogens. Despite these major differences, there were also common trends in the transitions between healthy controls, CLD and HCC patients. In blood, several mostly saturated triacylglycerides showed a continued increase in the trajectory towards HCC, accompanied by reduced levels of saturated free fatty acids and saturated lysophospatidylcholines. In contrast, the largest overlaps of lipid alterations that were found in both HCC tissue and blood comparisons were decreased levels of phosphatidylglycerols and sphingolipids. This study highlights the specific impact of HCC tumors on the circulating lipids. Such data may be used to target lipid metabolism for prevention, early detection and treatment of HCC in the background of viral-related CLD etiology.

## 1. Introduction

Hepatocellular carcinoma (HCC), the main type of liver cancer, is the fourth-leading cause of cancer related deaths worldwide [1,2]. HCC incidence and its main risk factors show notable disparities across countries [3,4]. In the United States, an increasing rate of HCC is associated with alcoholic liver disease, nonalcoholic fatty liver disease (NAFLD), metabolic syndrome and hepatitis C virus infection (HCV) [2,5], all of which are directly associated with dyslipidemia [5,6,7,8]. In Egypt and China, chronic liver disease (CLD) is the main risk factor of HCC [4,9] due to HCV and hepatitis B virus infection (HBV). Chronic liver disease (CLD) and its end stage liver cirrhosis are the direct causes of HCC. Clinically, the underlying CLD in HCC patients hampers early diagnosis due to overlapping manifestations and laboratory markers, such as serum alfa fetoprotein (AFP) and liver function tests [10,11]. Identifying potential tumorigenesis in CLD patients could provide a possible diagnostic clue and timely treatment to HCC patients. The liver has a central role in lipid metabolism. Hepatic lipid metabolism is linked to carbohydrate and protein metabolism to maintain normal body homeostasis and growth [12,13]. In hepatic diseases, the disturbance of liver functions is associated with aberrant lipid metabolism [14,15]. Alterations in lipid metabolism in HCC-related matrices is usually investigated by mass spectrometry based lipidomics [16]. Disturbed triacylglycerol (TG) and phosphatidylglycerol (PG) metabolism in HCC tissues from mixed etiologies has been reported [17,18,19,20], in addition to a decrease in the blood levels of cholesteryl esters (CE) and alterations in blood sphingomyelins and free fatty acids [16,20,21,22]. While HCC metabolic phenotypes have been studied independently of either blood or hepatic tissues [16], the association between tumor tissues and blood lipidomics in the same HCC patients has not been clearly determined yet [23,24]. We have previously shown that sugar alcohols are significantly upregulated in CLD and HCC whole blood and tissues compared to nontumor hepatic tissues [25]. We here report a follow-up study on aberrant lipid metabolism in blood of CLD and HCC from the same viral etiology patients compared to healthy control subjects. We also show how lipidomic changes detected in blood correspond to dyslipidemia in HCC tumor tissues compared to paired nontumor hepatic tissues.

## 2. Results

### 2.1. Untargeted Lipidomic Using UPLC MS-QTOF of Whole Blood and Liver Tissues of Hepatocellular Carcinoma (HCC) and Chronic Liver Disease (CLD) Patients and Healthy Control Subjects

Untargeted lipidomic profiling was conducted on the whole blood of 23 HCC, 15 CLD and 15 healthy control subjects. In addition, 46 hepatic tissues were resected from the same 23 HCC patients in pairs of HCC tumors and surrounding nonmalignant tissues. Demographic and biochemistry data are given in Appendix A, as also published previously [25]. Untargeted lipidomics using Agilent UPLC-QTOF MS (Santa, Clara, CA, USA) combined positive and negative electrospray ionization mode revealed 604 unique identified lipid metabolites from a total of 2498 mass-retention time deconvoluted lipid signals. The identified lipid metabolites were categorized into six lipid super classes comprised of a total of 16 subclasses: phospholipids; ceramides and sphingolipids; neutral glycerolipids; fatty acids; acylcarnitine; and sterols (Figure 1). Phospholipids present the most abundant lipid class detected in both blood and tissue samples with 58% of all identified species. Within this class, phosphatidylcholines (PC), plasmalogens and phosphatidylethanolamines (PE) were found as the dominant subclasses (Figure 1). The second most diverse lipid superclass, sphingolipids, was equally divided between sphingomyelin and ceramide derivatives (Figure 1). Neutral glycerolipids mainly consisted of triacylglycerols (TG), and to lesser extent, diacylglycerols (DG) and few monoacylglycerides (Figure 1). The 16 lipid subclasses were further categorized according to the numbers of carbon atoms and double bonds in their corresponding acyl chains. Such characterizations by acyl chain length and degree of unsaturation are important distinctions that reflect metabolism and biological function. A summary of the identified lipids organized by types of adduct, experimental mass and retention time is given in Appendix A for both blood and tissues.

### 2.2. Blood Lipidomic Profiles of HCC and CLD Patients and Healthy Control Subjects

To enable detailed statements on dysregulation of lipid metabolism, we constrained all statistical analyses to the set of 604 identified lipids, rejecting signals of unidentified lipids. We started our investigation by multivariate analysis to give an overview about the degree at which blood lipids were contributing to the differentiation of CLD and HCC patients and healthy control subjects (Figure 2). Such multivariate overview plots also assist in finding outlier samples and analytical bias. The raw total ion concentrations chart of all detected blood lipids showed that CLD patients tend to have higher lipid levels compared to healthy control subjects, while HCC patients showed the largest variance in total lipid levels (Figure 2a). Unsupervised principal component analysis (PCA, Figure 2b) showed that 27% of the overall variance (PCA vector 2) in individual blood lipids was sufficient to completely separate healthy controls from both CLD and HCC patients. Supervised multivariate statistics using sparse partial least squares discriminant analysis (sPLSDA) models (Figure 2c) yielded sPLSDA vector 2 that almost completely separated CLD from HCC patients, indicating a specific blood lipidomic signature for HCC subjects. Applying univariate statistical analysis using Kruskal–Wallis test revealed 524 significant (*p* value < 0.05, q FDR < 0.1) metabolites differentiating HCC, CLD and healthy control blood samples (Appendix A). We used lipid group averages with ward clustering to construct a hierarchical clustering heatmap that revealed four distinct lipid clusters (Figure 2d). Cluster A represented blood lipids with increased levels in CLD patients and decreased levels in HCC patients consisting of some lysophosphatidylcholines (LPC) and more lysophosphatidylethanolamine (LPE) species. In the same cluster A, saturated free fatty acids were found at largely decreased levels, indicated by blue bands, in both CLD and HCC patients compared to healthy control subjects. Cluster B comprised lipids with increased levels in both CLD and HCC groups compared to the healthy control group, for example, phosphatidylcholines (PC) and ceramides. Cluster B included 11 compounds with increased levels in the HCC group compared to CLD and healthy control groups, for example, PC (37:3) and PC (37:4), monounsaturated free fatty acids such as FA (19:1) and FA (16:1), and ceramides Cer (d34:0), Cer (d34:1) and Cer (d34:2). Cluster C represented lipids with increased levels in CLD patients compared to both the HCC and the healthy control group, mainly consisting of triacylglycerides (TG). Cluster D yielded lipids with increased levels in CLD patients compared to HCC subjects, while the healthy control group showed even lower lipid levels compared to either of the diseased groups. Sphingomyelins, phosphatidylglycerides (PG), phosphatidylethanolamines (PE) and diacylglycerides (DG) were the main constituents of cluster D. Appendix A details the complete panel for each cluster.

To obtain more details about the trajectory of metabolic differences from healthy controls to CLD patients and then to HCC subjects, we compared these subjects in pairwise multivariate statistics models (Appendix A), in addition to nonparametric univariate statistical analysis using the Mann–Whitney U test. 500 significant lipids were found at *p* < 0.05 and q FDR < 0.1 in the transition from healthy subjects to CLD patients, and 398 significant lipids were found when comparing HCC patients to CLD patients (Appendix A). Significantly altered metabolites were visualized in Volcano plots to indicate significance levels along with the direction and magnitude of change (Figure 3). Similarly to the overall trend in total blood lipid contents (Figure 2a), the detailed Volcano plot of individual lipids showed large increases in numerous blood lipids in CLD patients versus healthy control subjects, while in HCC versus CLD patients, most of the significantly altered blood lipid metabolites were down regulated (Figure 3).

Classic univariate statistics assumes independence of variables but ignores the biological fact of interdependence of metabolites. For example, if lipases released fatty acyl moieties from phosphatidylcholines, the resulting blood signatures would reveal increased levels of free fatty acids and lysophosphatidylcholines, along with decreased levels of PCs. To investigate such relatedness of dependent variables, set enrichment statistics are used in omics analyses. In metabolomics, lipid sets can be defined by relatedness of chemical structures using chemical set enrichment statistics in ChemRICH [26]. ChemRICH categorizes each compound to its compound class and calculates significance levels and fold-change of each class in a pair-wise comparison. ChemRICH plots detailed the very significant upregulation of almost all lipid classes in CLD patients versus healthy control subjects (Figure 4a), even after false-discovery rate correction (qFDR). Triacylglycerides were found as the most significantly upregulated lipid class at qFDR = 1.3 × 10^−56^, followed by PCs at qFDR = 3.3 × 10^−51^ and plasmalogens at qFDR = 1.8 × 10^−46^. In these plots, the bubble-size indicates the number of significant individual lipids in each set, and the color signifies the direction of change. Interestingly, around half of all free fatty acids and about 30% of all lysophosphatidylcholines were downregulated in CLD patients compared to healthy controls, indicated by the slightly purple colors of the enriched sets in the ChemRICH plots (Figure 4a, Appendix A). This finding suggests specific lipase and acyltransferase activities superseding the overall trend of upregulation of lipid metabolism in CLD patients. Following the trends indicated by heatmap clusters A, C and D (Figure 2d), the ChemRICH set enrichment plots showed most blood lipid classes are downregulated between HCC and CLD patients (Figure 4b). Several classes showed notable differences in individual lipid species, as indicated by purple bubble colors, with a range of individual triacylglycerols at increasing levels compared to CLD patients against the overall decreasing trends, and similar patterns for lysophosphatidylethanolamines (Figure 4b, Appendix A). Detailed statistics for each metabolite in each lipid class are given in Appendix A. Figure 4c gives the average intensity of each lipid class to visualize the overall trends of lipid regulation along disease progression from healthy controls to CLD and HCC patients. It becomes apparent that most lipid classes that were strongly upregulated in CLD patients still stayed above the levels of healthy controls, when compared to HCC patients. Exceptions from this trend were free fatty acids that decreased linearly from healthy control subjects to CLD and HCC patients, and lysophosphatidylethanolamines that decreased from CLD to HCC patients to levels that were even lower than in healthy controls (Figure 4c).

### 2.3. Lipidomic Profiles of Resected HCC Tumor Tissues Compared to Paired Nonmalignant Hepatic Tissues

Using all 604 identified lipids, we then investigated how lipid metabolism was altered in HCC tumor tissues compared to paired nontumor hepatic tissues from the same HCC patient and how such changes reflected changes observed in HCC and CLD patients. In most human cancers, lipids are upregulated compared to surrounding tissues to reflect the need for lipid membrane biosynthesis during cell division—for examples, take breast cancers [27] and lung cancer [28]. Surprisingly, in hepatocellular carcinoma, we found that overall changes were insufficient to yield clear lipidomic profiles to distinguish HCC tumors from paired nonmalignant samples using either unsupervised PCA or supervised sPLSDA multivariate statistical analyses (Appendix A). However, univariate statistics by nonparametric paired Wilcoxon signed-rank tests revealed 62 significantly different lipids at raw *p* < 0.05 (Appendix A). A volcano plot confirmed this finding (Appendix A). As explained above though, univariate statistics fail to take metabolic interdependencies into account. When using ChemRICH chemical set enrichment statistics, we discovered six lipid classes to be significantly altered in HCC tumors versus nontumor hepatic tissues (Figure 5a). Unlike other carcinomas, most lipid classes were downregulated in hepatocellular carcinoma, ranked from ceramides as the most significant class (*p* = 6.4 × 10^−16^ and qFDR= 1 × 10^−14^) to phosphatidylglycerols (*p* value = 3.7 × 10^−10^; qFDR= 3 × 10^−9^), phosphatidylcholines and plasmalogens (Figure 5a). Within each lipid class there were specific lipids that did not follow the overall trend in the class (Figure 5b). Specifically, free fatty acids and phosphatidylethanolamines had a roughly equal number of up and downregulated lipid species in their lipid classes (Figure 5a,b, Appendix A). Combined, these findings showed highly specific regulation of lipid classes and individual lipids in HCC tumor.

### 2.4. Differential Desaturation of Lipid Species Defines Metabotypes Both in Blood and in Resected Liver Tissues

Next, we used the ChemRICH tool to test whether subsets of lipid classes were altered with respect to the number of double bonds in their fatty acyl chains. Here we manually assigned all lipid species into three chemical sets with either 0, 1 or 2 and more double bonds for each lipid class (Figure 6).

Comparing blood of CLD patients to healthy controls, we found that lysoposphatidylcholines with 0 or 1 double bond were significantly downregulated (*p* < 2.5 × 10^−7^) but were upregulated with two or more double bonds (*p* = 1.7 × 10^−11^). Free saturated fatty acids were significantly downregulated (*p* = 0.0074) (Figure 6). These findings may suggest a higher activity of lysophosphatidylcholine acyltransferases to maintain levels of saturated phosphatidylcholines [29]. All other lipid species were found upregulated in CLD patients versus healthy controls, or insignificant. For blood of HCC versus CLD patients, we found levels of all lipid classes to be downregulated, regardless of the degree of unsaturation, except for monounsaturated free fatty acids. This specific subclass was significantly upregulated (*p* = 7.8 × 10^−4^), albeit at a low overall fold-change (Figure 6). In combination, the decrease of saturated fatty acids versus the increase in monounsaturated free fatty acids points to a higher activity of stearoyl CoA desaturase enzyme [30,31]. We also found significantly decreased levels for mono- and polyunsaturated ceramides (*p* = 2.8 × 10^−8^), but not for saturated ceramides (Figure 6).

In paired analyses of HCC tumor versus nontumor hepatic tissues, all ceramides were significantly downregulated, regardless of the degree of unsaturation (Figure 6). Similarly, all phosphatidylglycerols were found deregulated, but only mono- and polyunsaturated PGs reached ChemRICH significance levels. For phosphatidylcholines, only polyunsaturated PCs were found significantly downregulated, but not mono- and fully saturated PCs (Figure 6). The same trend was found for phosphatidylethanolamines (PE). Conversely, only saturated free fatty acids and saturated sphingomyelins were found downregulated in HCC tumors, but not their mono- or polyunsatured lipid species (Figure 6). Detailed results are given in Appendix A. Overall, any statistically significant lipid subclasses were found to be downregulated in HCC tumors compared to their surrounding nontumor hepatic resections. This finding may point to a depletion of constituent lipids that are already consumed by growing tumor cells. On the other hand, the difference between the degree of unsaturation within each lipid classes supports the notion of additional, specific enzymatic regulation.

### 2.5. Similarities and Differences in Lipid Metabolism the Transition from CLD to HCC Disease Patients

CLD is a direct risk factor of HCC. Hence, we were interested in finding blood-based compounds that are shared in the transition from healthy controls to CLD, and the continuing pathogenesis towards HCC etiology. Indeed, we found 21 lipid metabolites that showed such consistent alterations at statistical significance levels *p* < 0.05 (Figure 7b). Twelve of these lipids were found downregulated in blood of HCC patients compared to CLD subjects that were also downregulated between CLD subjects and healthy controls (Figure 7b). These compounds were mainly comprised of C16- and C18-acyl chain lysophosphatidylcholines, and the magnitude of effects was larger in the comparison of HCC/CLD than for CLD/healthy controls. Such lipids are products of specific phospholipases that act upon phosphatidylcholine membrane lipids. The fact that three different C16- and C18-LPC isomers were found to be significantly different points to alteration in the utilization of branched-chain alkyl groups that are typically found in food- or microbial sources. Supporting this idea, we found downregulation of odd-chain free fatty acids C17:0 and C19:0 that are also found in such exogenous sources. Furthermore, we found a specific lysocardiolipin downregulated containing three linoleic acyl groups (18:2) (Figure 7b), essential fatty acids that are retrieved from food, and again produced by the action of a specific lipase.

Conversely, we mostly found triacylglycerides to be consistently upregulated in the transition from healthy controls/CLD and from CLD/HCC patients (Figure 7b), and mostly consisting of saturated fatty acyl rests. Hence, these upregulations were not randomly distributed across all lipid classes, but the overall magnitudes of differences between HCC/CLD patients were smaller than for CLD/healthy controls. Few additional compounds were found consistently up regulated, including a single PC and a single PG. The biochemical interconnections between these lipid pathways are highlighted in Figure 7c. Overall, the increase in mainly saturated TGs along with the decrease in saturated LPCs and saturated free fatty acids may suggest a lower ratio of incorporation and use of lipids with highly saturated fatty acyl groups.

Next, we investigated if there were any common significant alterations in both blood and tissues from the same HCC patients (Figure 7a). Such data might give clues to common alterations of lipid metabolism homeostasis in the tumor itself and their imprints on circulatory lipids. Thirty-seven lipid metabolites showed such common significant alterations (Figure 7a). With the single exception of free fatty acid 20:3, all other significant and shared lipid changes were found as downregulated in both tumor/nontumor tissues and blood of HCC/CLD patients (Figure 7a). The largest set of downregulated lipids was phosphatidylglycerols, followed by ceramides, sphingomyelins and phosphatidylethanolamines. The decrease in biosynthesis of both ceramides and sphingomyelins was highly specific for saturated fatty acyl species (Figure 7a), again pointing to very specific differences in enzyme activities, whereas PG and PE lipids were mostly comprised of unsaturated fatty acyl residues. Such common fatty acyl-dysreglations are therefore clearly caused by specific regulatory circuits that ultimately may shed new insights into specific genes and enzymes that contribute to the pathogenesis of HCC from CLD.

## 3. Discussion

We previously reported increased sugar alcohols indicating dysregulated glucose metabolism in blood of chronic liver disease (CLD) patients and hepatocellular carcinoma (HCC) patients’ blood and tissues [25]. We suspected that aberrant glucose metabolism might also alter lipids because both pathways have common intermediate metabolites and signaling receptors [12]. We used whole blood samples to include membrane lipids of blood cells that represent overall lipid metabolism in a better way than just plasma lipids [32], because red blood cells contribute to informing us about differences between normal metabolism and disease states [33,34,35]. Importantly, we present herein the first study that focusds on a comparison of predominately HCV-induced hepatocellular carcinoma that combined tissue analyses with blood analyses, in relation to noncirrhotic chronic liver disease patients. Other studies, such as Lu et al.’s study [23], tested different cohorts, with only 1/50 HCC patients diagnosed with underlying HCV infection, and no CLD patient controls. It is therefore unsurprising that different lipidomic regulations were found. Even when only comparing our HCV-dominated cohort to healthy controls, discordant data compared to Lu et al. [23] were found, such as for triacylglycerides and plasmalogens. We found significant decreases in ceramides, whereas Lu et al. [23] did not report such differences. One of the few similarities between both studies were significantly decreased blood phosphatidylglycerides (PG) in HCC patients compared to healthy controls.

Under normal metabolic conditions, lipid metabolism homeostasis is controlled mainly by the liver and adipose tissue [36]. The liver is the major organ for lipid synthesis, oxidation and transport. It is responsible for the synthesis of apolipoproteins that are key for lipoprotein assembly and lipid transport [37]. In liver diseases, dysregulation of lipid synthesis, oxidation, storage, transport and chemokines has been frequently reported [12,38,39]. Chronic HCV and HBV viral hepatitis was reported to be associated with disturbed lipid metabolism [40,41]. HCV and HBV induce de novo lipogenesis that facilitates viral replication [40,42,43] while downregulating lipid oxidation that leads to steatosis [40,42,43]. The prevalence of HCV particles in the blood is associated with the formation of lipid droplets that contribute to viral pathogenesis and replication [44,45]. These studies support our findings of a significant upregulation of most lipid species in blood of viral-related CLD patients compared to healthy control subjects. It would be interesting to compare our result to data reported for different populations of patients with diverse liver disease phenotypes, such as nonalcoholic fatty liver disease (NAFLD), or nonalcoholic steatohepatitis (NASH). However, a comprehensive meta-analysis of blood lipidomic phenotypes in liver diseases is beyond of the scope of this work, also in regard to the small size of most cohorts reported to date.

Unlike colorectal [46], prostate [47], breast [48] and other types of cancer [49,50], in our study we found most blood lipids to be downregulated between HCC patients and their corresponding high-risk group of CLD patients. However, when comparing only HCC patients to healthy controls specific lipid classes were found up- or downregulated, as also reported for most other blood lipidomic studies in cancer. Therefore, differences in HCC-related lipidomic changes may be attributed to the different trajectories of HCC pathogenesis, either due to viral-related CLD or due to fatty liver-related CLD. Consequently, there is also a discrepancy in literature reports, depending on HCC-comorbidities and associated risk factors. Increased levels of serum triglycerides were reported in HCC patients without cirrhosis [51], and another report found decreased levels of triglycerides and cholesterol to be associated with progression in hepatic pathological conditions [52]. Our finding of downregulation of lipid species in HCC patients in both blood and liver might be explained by disruption of lipid homeostasis that balance lipoprotein synthesis along with consumption of lipids by the rapidly proliferating HCC tumor cells [53,54].

Similarly, cancer lipid profiles showed different findings in the degree of fatty acyl saturation in cancer patients [55,56,57]. We report decreased levels of circulatory saturated free fatty acids in the transition from healthy control to CLD and HCC patients that might be due to their role in activation of inflammatory cytokines [58]. Activation of the stearoyl-CoA desaturase (SCD) enzyme by the HCV replication machinery and in cancer cells may lead to an increased production of monounsaturated fatty acids concomitant to decreased levels of saturated fatty acids in viral-related CLD and HCC patients [59]. The increased incorporation of monounsaturated free fatty acids into phosphatidylcholines alters the membrane fluidity of cancer cells and prevents cancer related endoplasmic stress and apoptotic signaling in comparison to membrane with higher degrees of saturated fatty acyl groups [60]. Accordingly, we found decreased levels of lysophosphatidylcholines and increased levels of phosphatidylcholines in blood of CLD and HCC patients. We found eicosatrienoic acid (C20:3) increased in paired HCC tissue analysis but decreased in the comparison of blood levels of HCC to CLD patients. This fatty acid is an aberrant product in the arachidonate biosynthesis pathway [61], but similarly to our study, it was also found at decreased blood levels in liver disease patients [62]. It has previously also been reported as upregulate in colon cancer, possibly due to a decrease in lipid peroxidation and an increase in cell proliferation in tumor cells [63].

We report phosphatidylglycerols with decreased levels in both blood and tissue comparisons for HCC patients. Phosphatidylglycerols are phospholipid precursors of cardiolipins, which have an important role in mitochondrial wall function [63]. Alterations of phosphatidylglycerols were linked to hepatopathy and hepatic insulin resistance [64].

Ceramides and sphingomyelins belong to the lipid class of sphingolipids that are cell membrane components and signaling molecules controlling cell growth and apoptosis [65]. Ceramides and sphingomyelins are interconverted and are regulated by more than 28 enzymes that reflect the complexity of sphingolipids homeostasis [66]. Upregulation of serine palmitoyl transferase, a key enzyme of sphingolipids synthesis, has been linked to HCV replication and hepatic fibrosis process [67,68]. This report is in accordance with our finding of increased sphingolipids in blood of CLD patients compared to healthy control subjects. In cancer, increased levels of ceramides and sphingomyelins have been associated with less aggressive potential in breast [69], prostate [70] and gastric [71] cancers. Conversely, downregulated ceramides in colon cancer [72] and head and neck tumors [73] was found to be associated with a higher degree of tumor invasion and metastasis. In HCC tumor tissues, decreased levels of ceramides have been reported as mechanism to reduce the proapoptotic function of ceramides, possibly by an additional mechanism of tumor-promoting attraction of myeloid cells in the surrounding nontumor tissue [24]. Similarly to [24], we also found decreases in PE and ceramide levels in HCC tumors compared to nonmalignant surrounding tissues, yet in our report we also found decreases in sphingomyelins, polyunsaturated PCs, free fatty acids and importantly, phosphatidylglycerides that were not reported in [24]. Such reports should be combined in systematic meta-analysis studies and extended to larger cohorts to validate overall findings, including patient stratification by sex, viral infection status and other criteria, such as cirrhosis. We performed correlation analyses of blood metabolites of HCC patients against cancer stage, sex, number of focal lesions, size of focal lesion, AFP levels and current HBV and HCV infections. Perhaps due to the small size of the study, none of these correlations provided a strong and significant relation between lipid species and any of the given biochemical data.

## 4. Materials and Methods

### 4.1. Participants and Collection of Clinical Samples

Liver tissues and blood samples were obtained from 53 participants recruited from National Liver Institute, Egypt; 23 whole blood and pairs of tumors and nontumor liver tissue samples were obtained from HCC patients; 15 whole blood samples obtained from CLD patients; and another 15 whole blood samples obtained from age and gender matched healthy control subjects were also tested. The HCV infected patients in this study are genotype 4, as it is the most prevalent genotype in Egyptian HCC and CLD patients, and healthy control subjects included in this study were described in a previous study [25]. Each participant provided written informed consent. The study protocol (code 00185/2019) was approved by the ethical committee of the National Liver Institute at Menoufia University (NLI IRB00003413), Menoufia, Egypt in December 2019.

Initial diagnosis of HCC was based on serum AFP level > 200 ng/mL, ultrasound and triphasic spiral computed tomography imaging for focal lesions. This diagnosis was later confirmed histopathologically after surgical resection for all HCC cases. All CLD patients were diagnosed by PCR testing for hepatitis C virus and hepatitis B virus infection and validated clinically as positive chronic infection for more than 6 months. All CLD subjects were examined by diagnostic ultrasonography to exclude cirrhosis. All blood samples underwent liver function investigations: Aspartate transaminase (AST), alanine transaminase (ALT) and alpha-fetoprotein (AFP) were measured using a Beckman Coulter (Synchron CX 9 ALX) Clinical Auto analyzer (Beckman Instruments, Fullerton, CA, USA). Platelet count and Hemoglobin were analyzed by a Coulter Counter T660 (Coulter Electronics, Hielaeh, FL, USA). International normalized ratio (INR) spectra were obtained from prothrombin time measured by an STA-Stago Compact CT auto analyzer using reagents provided by Dade–Behring. Blood samples and tissue specimens were stored at −80 °C until the time of analysis. Samples were shipped on dry ice to the West Coast Metabolomics Center at UC Davis. Clinical and biochemistry characteristics of the cohort and HCC tissue characteristics are provided in Appendix A.

### 4.2. Sample Pretreatment

About 4 mg of frozen liver tissues were weighted and homogenized using stainless-steel grinding balls in GenoGrinder 2010 (Spex SamplePrep, Metuchen, NJ, USA) for 2 min at 1350 Hz. Homogenized tissue samples were subsequently used for metabolomic analyses in the same way as plasma samples.

Whole blood samples were aliquoted into 1.5 mL Eppendorf tubes, stored at −80 °C and shipped on dry ice. After thawing for 30 min at room temperature, samples were centrifuged for 30 min at 14,000× *g*; 20 µL of the supernatant was used for extraction as described [74]; 225 µL of cold methanol (MeOH) containing a mixture of 27 internal standards (Appendix A) was added to the samples and then vortexed for 10 s; 750 µL of methyl tertiary-butyl ether (MTBE) was added and samples were vortexed for 10 s and shaken for 5 min at 4 °C. Next, 188 µL water was added, vortexed and centrifuged for 2 min at 14,000 rcf. Two 350 µL aliquots from the nonpolar layer were prepared. One aliquot was stored at −20 °C as a backup and the other was evaporated to dry in a Speed Vac. Dried extracts were resuspended by 60 µL of a mixture of methanol/toluene (9:1, *v/v*) containing an internal standard [12-[(cyclohexylamino) carbonyl]amino]-dodecanoic acid (CUDA)] as a quality control. Method blanks and pooled human plasma (BioIVT; Westbury, NY, USA) were extracted and analyzed alongside with the study samples.

### 4.3. Lipidomic Data Acquisition and Data Processing

Liquid chromatography used a Waters Acquity UPLC CSH C18 column (100 mm × 2.1 mm, 1.7 µm particle size) coupled to an Acquity UPLC CSH C18 VanGuard precolumn (5 × 2.1 mm; 1.7 µm) (Waters, Milford, MA, USA) with mobile phases of 60:40 acetonitrile/water (A) and 90:10 isopropanol/acetonitrile (B). Mobile phases were buffered with 10 mM ammonium formate and 0.1% formic acid for positive electrospray ionization (ESI) analysis, and 10 mM ammonium acetate and 0.1% acetic acid for negative ESI analysis. Mobile phase gradient started with 85% (A) at 0 min; changed to 70% (A) from 0 to 2 min, 52% (A) from 2 to 2.5 min, 18% (A) from 2.5 to 11 min, 1% (A) from 11 to 11.5 min, 1% (A) from 11.5 to 12 min; and finally went to 85% from 12 to 12.1 min and continued as 85% (A) from 12.1 to 15 min. Sample temperature was maintained at 4 °C in the autosampler. Samples were reconstituted in LC starting buffers; 2 µL samples were injected to the column for positive ESI analyses, 5 µL samples for negative ESI analyses. For extracted liver samples, 1 µL was injected in positive ESI mode and 3 µL in negative ESI mode.

Mass spectrometric detection of lipids was performed using an Agilent 6530 Quadrupole Time of Flight Mass Spectrometer (QTOF MS) system (Santa Clara, CA, USA) in positive ESI mode, and an Agilent 6550 QTOF MS system (Santa Clara, CA, USA) for analysis of negative ESI mode. Samples were analyzed in a randomized order with interspersed method blanks and plasma quality control samples.

Data were processed using open-source software MS-DIAL version 3.98 [75]. MS-DIAL software allowed baseline correction, peak detection, alignment, gap filling and adduct identification for raw data. MS-DIAL performed accurate mass/retention time (m/z-RT) match to an in-house m/z-RT library, while MS/MS was matched to library spectra from the Mass Bank of North America (MoNA), NIST17, and LipidBlast [76]. All method blanks detected features were removed from further investigation. Mass Spectral Feature List Optimizer (MS-FLO) [77] was used to investigate duplicate peaks, isotopes and adducts. Peak height was used as mass spectral intensity at a specific retention time for each annotated lipid.

### 4.4. Statistical Analysis

Differential analysis of biochemical and clinical data was performed using t-tests and chi-square statistical tests (Appendix A). For lipidomic data, nonparametric univariate and multivariate statistical tests were used to test for significantly altered individual lipid metabolites and lipid classes between the study groups. The statistical tests were applied to nonnormalized data generated after MS-DIAL processing. The Kruskal–Wallis test was used to detect significant altered features between the three study groups and the Mann–Whitney U test was used for pairwise comparisons in blood. The Wilcoxon signed rank test was used to investigate data from paired liver tissue samples. From all comparisons, *p*-values less than 0.05 and adjusted *p*-value using the Benjamini-Hochberg’s false discovery rate (FDR) q < 0.10 were considered significant. Unsupervised multivariate analyses were used mainly for quality control, such as presence of outliers or any analytical bias. Sparse partial least squares—discrimination analysis (sPLS–DA), a multivariate statistical method, was applied to reflect the comparison of a few samples against a high number of variables [78]. Heat map hierarchical clustering was performed to show a graphical summary of differentially altered metabolites, using a clustering Euclidian for distance measurement allow assorting data by similarity patterns. Volcano plots showed statistically significant identified lipid metabolites and fold change (FC) represented as log FC. Enrichment analysis to point chemical classes that were significantly differentiated across blood and tissue study groups was performed by ChemRICH software [26]. ChemRICH is a web interface that utilizes chemical ontologies and structure similarity to map metabolites to their metabolic modules. Venn diagram is a graphical organizer summarizing shared lipidomic data across the study groups. Venn diagram was performed using Venny 2.1.0 [79]. Statistical analyses were calculated using R software version 3.5.3 (R Foundation for Statistical Computing, Vienna, Austria), MetaboAnalystR 3.0 [80] and lipidr [81].

## 5. Conclusions

The liver has a central role in lipid metabolism. Consequently, our lipidomics analyses showed statistically significant differences between blood of healthy controls, CLD and HCC patients. Interestingly, most lipids that were found upregulated in blood of CLD patients were reversed in HCC patients, showing differences in the etiology of CLD on the one hand and HCC on the other. Similarly, we found many lipids to be downregulated in HCC tumor tissues compared to paired nontumor hepatic tissues. Interestingly, several lipid changes showed a continued trend in the transition from blood of healthy controls to CLD and HCC patients, for example, decreased levels of saturated free fatty acids and saturated lysoposphatidylcholines along with upregulation of saturated triacylglycerides. Once these findings are replicated in additional cohorts, the risk for viral-related CLD patients to develop HCC may be monitored by blood-based biomarker analysis. In addition, our findings of differential regulation of sphingolipids and phosphatidylglycerols in blood and tissues of HCC patients may open new opportunites for using the corresponding enzymes as therapeutic targets. Overall, we here present the first study comparing both blood and hepatic tissue-based lipidomics of viral-related HCC to its direct risk factor CLD.

## Figures and Tables

**Figure 1 cancers-13-00088-f001:**
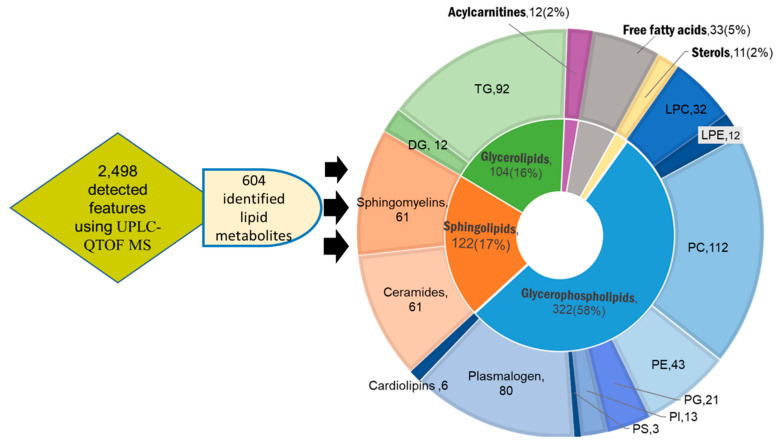
Overview of lipids identified in blood and tissue samples of hepatocellular carcinoma (HCC), chronic liver disease (CLD) and healthy control whole blood subjects using UPLC–QTOF MS. Inner circle: categorization into 6 lipid super classes: purple, acylcarnitines; grey, free fatty acids; yellow, sterols; blue, glycerophospholipids; orange, sphingolipids; green, glycerolipids. Outer circle: categorization into 16 subclasses matching the colors of the related superclasses. Each lipid class is presented with the number of its members and percentage of the total identified lipid metabolites.

**Figure 2 cancers-13-00088-f002:**
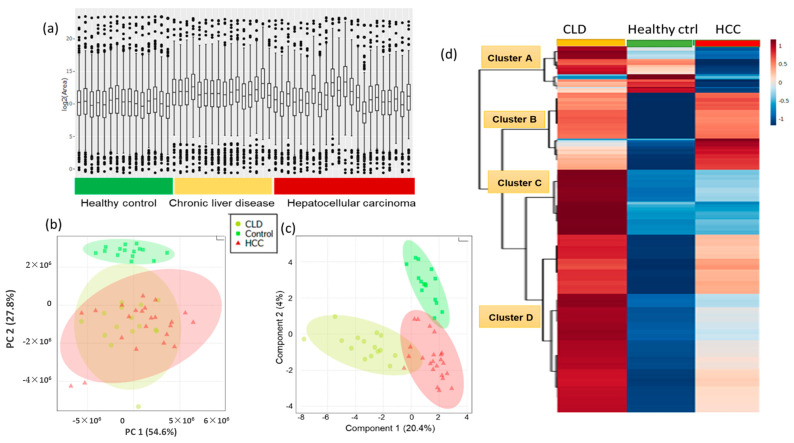
Multivariate and univariate statistical analyses of identified blood lipids in whole in hepatocellular carcinoma (HCC) and chronic liver disease (CLD) patients, and healthy control subjects. (**a**) Distribution of total ion concentrations (TIC) of all lipids for each subject, organized by subject groups. (**b**) Unsupervised principal components analysis (PCA) separating lipid profiles of HCC, CLD and healthy control groups. (**c**) Supervised sparse partial least squares discriminant analysis (sPLSDA) separating lipid profiles of HCC, CLD and healthy control groups. (**d**) Hierarchical clustering heatmap of 524 significant lipids after univariate Kruskal–Wallis tests (*p* < 0.05 with qFDR < 0.1). Columns represent lipidomic averages of each subject group. Levels of individual lipid average intensities colored by auto-scaling from high (red) to blue (low) levels.

**Figure 3 cancers-13-00088-f003:**
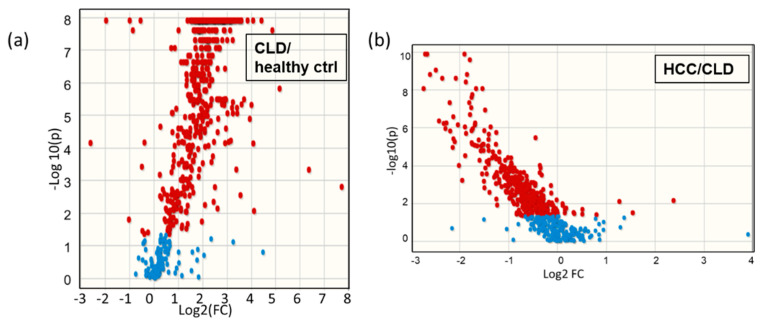
Volcano plots of the pairwise comparison blood lipids. (**a**) Chronic liver disease patients (CLD) versus healthy controls; (**b**) HCC versus CLD subjects. FC indicates fold changes of lipid averages in the two group comparisons.

**Figure 4 cancers-13-00088-f004:**
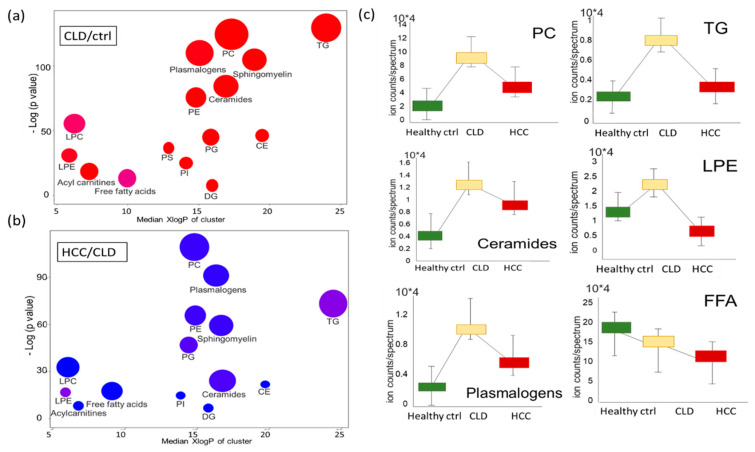
Dysregulated blood lipid metabolism of healthy control subjects and chronic liver disease (CLD) and HCC patients. (**a**) Chemical set enrichment statistics (ChemRICH) comparing CLD patients to healthy control subjects. (**b**) ChemRICH plot comparing HCC to CLD patients. Bubble colors range from red (increased) to blue (decreased), with purple colors indicating both increased and decreased lipid species in lipid classes. Bubble size represents the number of compounds per lipid class. (**c**) Box plots of average blood lipid intensities (normalized peak heights) with standard deviations, indicating differences between healthy controls and CLD and HCC groups. Abbreviations: PC, phosphatidylcholines; PE, phosphatidylethanolamines; PG, phosphatidylglycerols; PS, phosphatidylserines; PI, phosphatidylinositols; CE, cholesteryl esters; LPC, lysophosphatidylcholines; LPE, lysophoaphatidylethanolamines; LPC, lysophosphatidylcholines; FFA, free fatty acids; DG, diacylglycerols; TG, triacylglycerols.

**Figure 5 cancers-13-00088-f005:**
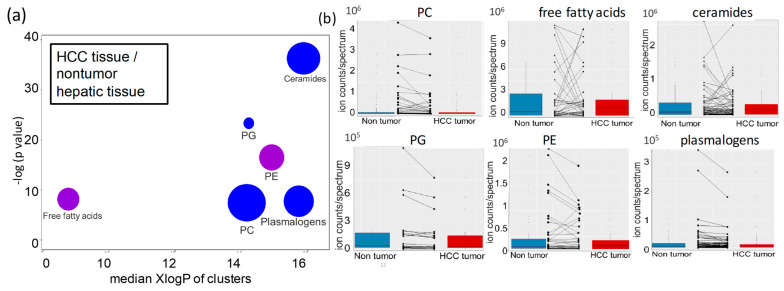
Lipid set enrichment statistics in HCC hepatic tissues versus nontumor hepatic tissues (**a**) ChemRICH plot. Bubble colors range from blue (decreased) to purple, indicating increased and decreased lipid species in lipid classes. Bubble size represents the number of compounds per lipid class. Lipid classes are ordered by median lipophilicity (XlogP) along the x-axis. (**b**) Boxplots of paired comparisons of significant lipid classes in HCC tumors versus nontumor hepatic tissues, including directions of change for individual lipid species in these classes. Abbreviations as in Figure 1.

**Figure 6 cancers-13-00088-f006:**
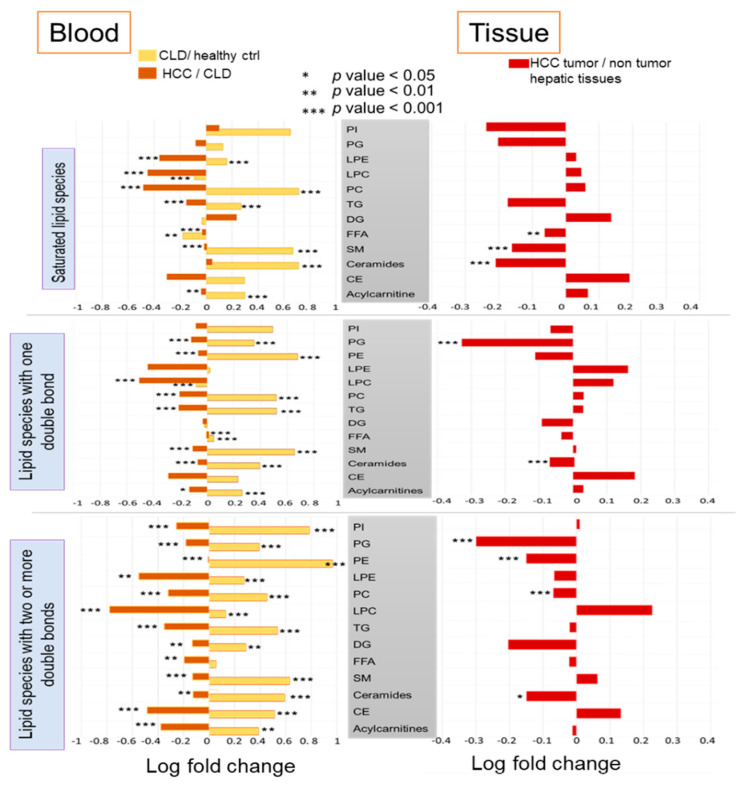
Differential lipid set enrichment statistics according to the level of desaturation in fatty acyl groups for each lipid class. Left panel: blood lipids. Right panel: HCC tumor lipids versus paired nontumor hepatic tissues.

**Figure 7 cancers-13-00088-f007:**
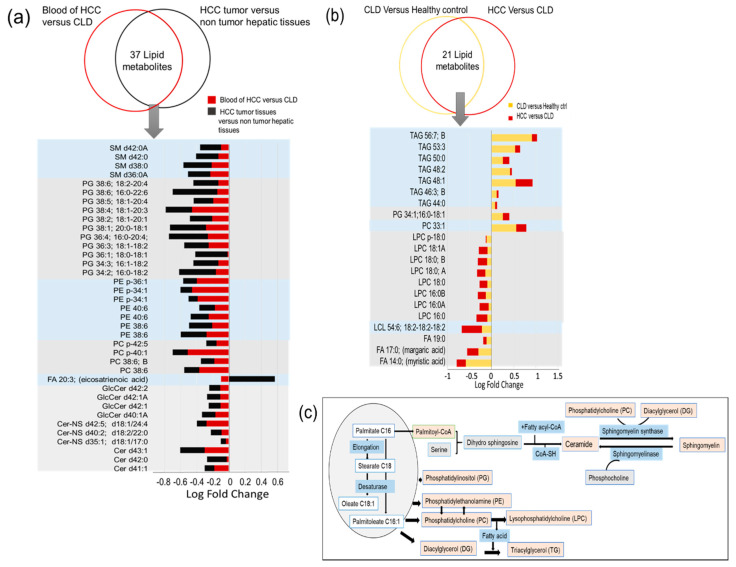
Lipid species that were found significantly dysregulated in the transition of CLD to HCC. (**a**) Common and shared significantly altered lipids in blood and tissues of the same HCC patients. (**b**) Significantly altered metabolites that share the same direction of alteration in the blood of HCC versus CLD patients and CLD patients versus healthy control subjects. (**c**) Metabolic pathways for synthesis of major lipid classes in the human body.

## Data Availability

Data are freely available at the MetabolomicsWorkbench.org.

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
