# Peer review of "Remodeling Lipids in the Transition from Chronic Liver Disease to Hepatocellular Carcinoma"

_cancers, 2020, doi:10.3390/cancers13010088_

Round 1
Reviewer 1 Report
In the manuscript of Ismail et al, authors characterized the lipid content in blood but also in hepatic tissues and tumors from patients with chronic liver disease (CLD) and HCC in the context of viral infection. The topic of this paper is interesting. Although several studies have already identified (but separately) the lipids deregulated in the liver tissues/tumors or blood from HCC/CLD patients, this study provides an integrated analysis of all these alterations in these different tissues. This study may therefore uncover important lipids, which may contribute to the transition CLD-HCC and thus may suggest the targeting of specific metabolic pathways for therapeutic purpose. The manuscript is well written and organized and the methodologies are properly described. However, some improvements are required and thus it is recommended that the authors follow these specific comments:
Major comments
1-The lipids identified in this study and deregulated in CLD and HCC should be compared with previously published studies in order to highlight the novelty of this study. Do we have other lipids, which were not identified before? What are the common one? Is there any difference with other etiologies (NASH, ALD)? Plasmalogens, ceramides and lysophosphatitic acids were previously reported in other studies (PMID: 29435160).
2-Authors found that ceramides, phosphatidylglycerols, phosphatidylcholines and plasmalogens are downregulated in HCC as compared to surrounding non-tumoral tissues. Such effects have previously been described in a mixed cohort (e.g, PMID: 31040908: Alcohol, viral infections, others), thus suggesting that this lipid signature is not specific to HCC in the context of viral infection. Indeed, the low level of ceramides protects HCC cells from oxidative stress and apoptosis (e.g, PMID: 3205803). Therefore, this aspect of the study is more confirming what others have previously identified. Authors should stress more the differences between their study and the others, especially the differences with the double bonds in their fatty acyl chains. Finally, authors should attempt to explain the role of these specific lipid classes (with different amount of double bonds) in CLD and the progression toward HCC. Are they studies documenting the role of these classes in inflammation or processes contributing to the progression toward HCC?
3-The figure 6 is very interesting but should better highlight the lipid species commonly deregulated between healthy Ctl/CLD and preserved with HCC, because they are likely the ones involved in the progression toward HCC. A small table and/or a graphical representation should be provided to better highlight this message. Moreover, these deregulated species may contribute also to the development of HCC in non-cirrhotic patient in a context of NAFLD (maybe to discuss in the discussion section).
4-Authors have a mixed cohort in the CLD group, with patients infected by HCV and others by HBV. It is therefore unclear, whether, the differences observed in lipid species in the blood or in the tissues in CLD vs CTL are coming from HCV, HBV or both groups. What is the degree of similarities of lipid species between HCV and HBV patients? Authors should perform additional analyses to stratify these patients. Moreover, authors should provide more information about the HCV genotypes in the patients (3a, 1b?), because they affect differently lipid metabolism.
Minor comments
1-It could be interesting to see if the lipid entities identified in the blood of HCC patients correlate with some clinical feature: prognosis, tumoral grade etc. Indeed, these entities could be used as biomarkers in liquid biopsies.
2-The potential alteration of lipids with anti-HCV drugs (sofosbuvir) should be discussed in the discussion section. Do we know if patients treated with these drugs have still a deregulation of lipid metabolism in the liver and thus of lipid species in the blood?
Author Response
Reviewer 01
In the manuscript of Ismail et al, authors characterized the lipid content in blood but also in hepatic tissues and tumors from patients with chronic liver disease (CLD) and HCC in the context of viral infection. The topic of this paper is interesting. Although several studies have already identified (but separately) the lipids deregulated in the liver tissues/tumors or blood from HCC/CLD patients, this study provides an integrated analysis of all these alterations in these different tissues. This study may therefore uncover important lipids, which may contribute to the transition CLD-HCC and thus may suggest the targeting of specific metabolic pathways for therapeutic purpose. The manuscript is well written and organized and the methodologies are properly described. However, some improvements are required and thus it is recommended that the authors follow these specific comments:
Major comments
- The lipids identified in this study and deregulated in CLD and HCC should be compared with previously published studies in order to highlight the novelty of this study. Do we have other lipids, which were not identified before? What are the common one? Is there any difference with other etiologies (NASH, ALD)? Plasmalogens, ceramides and lysophosphatitic acids were previously reported in other studies (PMID: 29435160).
Response: We here focus on the etiology for HCC by HCV infection, specifically for the Egyptian variant of genotype4 HCV. It is beyond the scope of this paper to comprehensively review lipid profiles against other diseases, but we added notes to a few comparison studies in the discussion section. Specifically, the study PMID 29435160 by Lu et al (Oncotarget 2018) is indeed quite interesting, as it at least compared liver resections and blood, but studied a different patient cohort than ours. Only 1/50 patients in the Lu et al study was positive for HCV virus, while in our study, the majority of patients were positive for HCV when presenting to the clinic. The Lu et al study did not include other patients with CLD. In comparison to healthy controls, Lu et al found saturated blood triacylglycerides down-regulated, whereas we did not find such comparisons particularly interesting: the clinical question is the differentiation of HCC patients to CLD patients, not to healthy controls. Nevertheless, in our HCV-dominated cohort, saturated blood triacylglycerides were still higher in HCC than in healthy controls (Figure 4, 6). Similarly, in our HCV-dominated cohort, blood plasmalogens were increased compared to healthy controls (Lu et al: down), and free fatty acids were not even examined by Lu et al. In paired tumor/non-tumor tissues, Lu et al found increased triacylglycerides whereas in our HCV-dominated tumors, triacylglycerides were found decreased (but not significant). We found significant decreases in ceramides, Lu et al did not report such finding. One of the few similarities were significantly decreased phosphatidylglycerides (PG) which we also now added to the discussion section (lines 319-328).
- Authors found that ceramides, phosphatidylglycerols, phosphatidylcholines and plasmalogens are downregulated in HCC as compared to surrounding non-tumoral tissues. Such effects have previously been described in a mixed cohort (e.g, PMID: 31040908), thus suggesting that this lipid signature is not specific to HCC in the context of viral infection. Indeed, the low level of ceramides protects HCC cells from oxidative stress and apoptosis (e.g, PMID: 3205803). Therefore, this aspect of the study is more confirming what others have previously identified. Authors should stress more the differences between their study and the others, especially the differences with the double bonds in their fatty acyl chains. Finally, authors should attempt to explain the role of these specific lipid classes (with different amount of double bonds) in CLD and the progression toward HCC. Are they studies documenting the role of these classes in inflammation or processes contributing to the progression toward HCC?
Response: We agree that Cotte et al. (PMID 31040908) gave an interesting report, yet, on a different cohort. First of all, our CLD patient controls were not cirrhotic (let alone alcohol-related), unlike for Cotte et al. Secondly, Cotte et al. did not study tumor resections versus surrounding tissues, but only plasma. Yet, Cotte et al. discussed findings of Krautbauer et al. that we also cited. In the discussion section, we now highlight commonalities in tissue alterations to tumors with decreases in PE and ceramides. Yet, in our report we also found decreases in sphingomyelins, polyunsaturated PCs, free fatty acids, and, importantly, phosphatidylglycerides. We make these commonalities and differences now clearer. Nevertheless, due to the small size of the studies, the high diversity of patient characteristics, lack of clear intervention-based differences, we do not want to engage in mechanistic discussions such as a potential role of ceramides in protective actions against oxidative stress and modulating apoptosis. The role of specific lipids in disease etiologies are best studied in cells and animal models, for which human cohort studies can only give the foundation of onset and progression of diseases. We restrict ourselves to reporting findings and similarities but would not like to stretch interpretations beyond what data show. As we did not have data on inflammatory status between different groups of CLD or HCC patients, we cannot comment on that possibility.
- The figure 6 is very interesting but should better highlight the lipid species commonly deregulated between healthy Ctl/CLD and preserved with HCC, because they are likely the ones involved in the progression toward HCC. A small table and/or a graphical representation should be provided to better highlight this message. Moreover, these deregulated species may contribute also to the development of HCC in non-cirrhotic patient in a context of NAFLD (maybe to discuss in the discussion section).
Response: Figure 7b highlights the lipid species that are common between these study groups. Again, we would like to present the data here, but we cannot distinguish cause and effect, i.e. we cannot tell if changes in specific lipids from healthy controls to CLD to (NAFLD) HCC might be involved mechanistically in the development of HCC, or if these lipids are consequences (instead of actors) of underlying mechanistic dysregulations.
- Authors have a mixed cohort in the CLD group, with patients infected by HCV and others by HBV. It is therefore unclear, whether, the differences observed in lipid species in the blood or in the tissues in CLD vs CTL are coming from HCV, HBV or both groups. What is the degree of similarities of lipid species between HCV and HBV patients? Authors should perform additional analyses to stratify these patients. Moreover, authors should provide more information about the HCV genotypes in the patients (3a, 1b?), because they affect differently lipid metabolism.
Response: We agree to this statement and added a confirmatory comment to the discussion section. However, we feel that the cohorts were too small to distinguish these groups, especially as one would also need to stratify the groups by sex. HCC patients are clearly dominated by men, and larger studies are needed to distinguish specific lipid phenotypes by sex, by viral infection status and other criteria such as cirrhosis. Thank you for the comment on HCV genotypes which we now clarified as genotype 4.
Minor comments
- It could be interesting to see if the lipid entities identified in the blood of HCC patients correlate with some clinical feature: prognosis, tumoral grade Indeed, these entities could be used as biomarkers in liquid biopsies.
Response: This is a small cross-sectional study that did not have additional follow up on survival times or remission status. We followed the suggestion by the reviewer and performed correlation analyses of HCC patients’ blood metabolites against cancer stage (early means stage I and II while late stage III and IV), gender, number of focal lesions, size of focal lesion, alfa fetoprotein (AFP) levels, current HBV and HCV infections. Perhaps due to the small size of the study, none of these correlations could provide a strong and significant relation between lipid species and any of the given biochemical data. We give this comment now in the discussion section.
- The potential alteration of lipids with anti-HCV drugs (sofosbuvir) should be discussed in the discussion section. Do we know if patients treated with these drugs have still a deregulation of lipid metabolism in the liver and thus of lipid species in the blood?
Response: In Egypt, treatment with anti-HCV drugs is stopped once HCC is suspected in HCV infected patients due to the possible effect of antiviral treatment on the HCC progression (PMID: 29962815). But antiviral treatment continues to take in HCC patients associated with HBV infection. 2/23 HCC patients in our study were HBV positive and under treatment at the time of the study, a number that is too small to be studied statistically. At least, profiles of those two patients did not show as outliers in multivariate statistical analyses. In our study, CLD patients started anti-HCV treatment after samples had been obtained.
Reviewer 2 Report
The article: “Remodeling Lipids in the Transition from Chronic Liver Disease to Hepatocellular Carcinoma” by Israa T. Ismail et al. describes the lipid profile in blood of controls, patients with chronic liver disease and HCC patients. Moreover, the tumor lipidome was characterized.
„While HCC metabolic phenotypes have been studied independently on either blood or hepatic tissues [16], only a single previous study reported on lipid changes in both serum and HCC hepatic tissues [23].“ This is not correct and there is at least one other study DOI: 10.1016/j.bbalip.2016.08.014
Chronic HCV is associated with fatty liver and a decrease in serum lipoproteins. “Based on the result of this large scale community study, HCV viremia appears to be associated with lower serum cholesterol and triglyceride levels which implies that HCV itself might play a significant role on serum lipid profile of patients with chronic HCV infection. “(DOI: 10.1016/j.jhep.2008.03.016 and DOI: 10.1097/MEG.0b013e32833de92c). Here, levels of CE and other lipids were all increased in the HCV patients. This is an unexpected finding. Was LDL also induced in the HCV patients? Authors should include levels of lipoproteins in table S1.
Moreover, it is essential to have any idea about liver disease severity. Child-Pugh score, MELD score or histological evaluation of liver fibrosis has to be included in table S1.
Most lipids decline in HCC. CLD patients did not have liver cirrhosis. In case HCC patients had liver cirrhosis how do the authors discriminate lipid changes related to cirrhosis and lipid changes related to HCC?
Figure 1, please use identical font sizes. Please explain the numbers and the %. What are the purple lipids?
Figure 5, no red bubbles in this figure.
“we found decreased levels of all lipid classes to be downregulated” “our lipidomics analyses showed significant differentiation „please correct sentences like these.
Please include HCV genotypes because viral genotype is related to liver steatosis.
Grammatikos et al. published that distinct sphingolipid species are induced in HCC, this study has to be discussed.
How many patients and controls had liver steatosis?
Li et al. (doi: 10.18632/oncotarget.23494) could not identify significant correlations between hepatic and serum lipids. They postulated “that hepatic and serum lipid signatures of HCC have to be considered as mostly independent, and the results imply potential roles of PEp species, particularly PEp (36:4) and (40:6), as serum biomarkers for HCC diagnosis and progression.” This paper should be discussed.
A recent work suggested that the CE/free cholesterol ratio is specifically induced in HCC. Was this also found in the present cohort?
Author Response
Reviewer 02
The article: “Remodeling Lipids in the Transition from Chronic Liver Disease to Hepatocellular Carcinoma” by Israa T. Ismail et al. describes the lipid profile in blood of controls, patients with chronic liver disease and HCC patients. Moreover, the tumor lipidome was characterized.
„While HCC metabolic phenotypes have been studied independently on either blood or hepatic tissues [16], only a single previous study reported on lipid changes in both serum and HCC hepatic tissues [23].“ This is not correct and there is at least one other study DOI: 10.1016/j.bbalip.2016.08.014
Response: The reviewer is correct, and we integrated this paper into our report. In the discussion section, we now highlight commonalities with Krautbauer et al. (2016) in tissue alterations to tumors with decreases in PE and ceramides. Yet, in our report we also found decreases in sphingomyelins, polyunsaturated PCs, free fatty acids, and, importantly, phosphatidylglycerides. We make these commonalities and differences now clearer. Importantly, we also state that each cohort had specific differences in patient characteristics. For example, in Krautbauer et al (2016), half of the patients had no fatty liver, whereas in Egypt, fatty liver disease is much more highly prevalent in both male and female populations. Similarly, our HCC patients were all cirrhotic (Child-Pugh grade A) whereas Krautbauer et al did not report cirrhosis. We added this information now to the supplement.
Chronic HCV is associated with fatty liver and a decrease in serum lipoproteins. “Based on the result of this large scale community study, HCV viremia appears to be associated with lower serum cholesterol and triglyceride levels which implies that HCV itself might play a significant role on serum lipid profile of patients with chronic HCV infection. “(DOI: 10.1016/j.jhep.2008.03.016 and DOI: 10.1097/MEG.0b013e32833de92c). Here, levels of CE and other lipids were all increased in the HCV patients. This is an unexpected finding. Was LDL also induced in the HCV patients? Authors should include levels of lipoproteins in table S1.
Response: Chronic HCV patients were recruited from Egypt where HCV genotype 4 is predominant and representing about 91% of HCV infections (doi: 10.3748/wjg.v24.i38.4330 and doi.org/10.1111/apt.12551.) An earlier study compared lipid profiles before and after direct acting antiviral drugs in chronic hepatitis C Egyptian patients with genotype 4; this study confirmed our results with increase in cholesterol and other lipid markers in CLD patients compared to heathy controls. This upregulation of lipids in chronic HCV infection is found to be improved by treatment (doi.org/10.1080/19932820.2018.1435124). For the patients in our study, LDL levels were not available to the authors due to restrictions for access to patient records. We added the genotype 4 information to the subjects sections.
Moreover, it is essential to have any idea about liver disease severity. Child-Pugh score, MELD score or histological evaluation of liver fibrosis has to be included in table S1.
Response: We added the Child-Pugh score to the subject supplementary table. MELD scores are only applied to advanced cases with severe liver failure, which did not apply to our patients. For HCC patients, histology confirmed tumors but for CLD patients, not liver biopsies were performed as this would be an undue invasive procedure (doi: 10.3748/wjg.v18.i23.2988).
Most lipids decline in HCC. CLD patients did not have liver cirrhosis. In case HCC patients had liver cirrhosis how do the authors discriminate lipid changes related to cirrhosis and lipid changes related to HCC?
Response: Chronic HCV patients in our study were examined by ultrasonography and liver function tests e.g. bilirubin, albumin, and prothrombin time to exclude cirrhosis. HCC patients underwent surgical resection for focal lesions and were exclusively rated Child-Pugh A. We now added this information to table S1.
Figure 1, please use identical font sizes. Please explain the numbers and the %. What are the purple lipids?
Response: We corrected this font sizes in the sunburst diagram. The purple section of the sunburst diagram relates to acylcarnitines and has been clarified. The number of compounds in each lipid class is given in relation to the total identified lipid compounds and presented as percentage, as it is common in the lipid literature. We modified the legend accordingly.
Figure 5, no red bubbles in this figure.
Response: Corrected.
“we found decreased levels of all lipid classes to be downregulated” “our lipidomics analyses showed significant differentiation „please correct sentences like these.
Response: Corrected.
Please include HCV genotypes because viral genotype is related to liver steatosis.
Response: Corrected and added to the subject and methods section.
Grammatikos et al. published that distinct sphingolipid species are induced in HCC, this study has to be discussed.
Response: In one of Grammatikos et al. studies (doi.org/10.1371/journal.pone.0138130), serum samples from cirrhotic patients were analyzed for sphingolipids together with follow up samples with appearance of complication such as ascites, encephalopathy and HCC. The Grammatikos study focused on the carbon length of ceramides in association with cirrhosis and its complication. In contrary, our study did not include cirrhotic patients, and we emphasized results on the number of double bonds in the side chain of lipid metabolites. Due to the large difference in patient cohort characteristics, these two studies cannot be reasonably compared.
How many patients and controls had liver steatosis?
Response: Diagnosis of steatosis is established by liver biopsy which was not performed in our CLD patients, let alone for controls. Interestingly, steatosis has a relatively high prevalence in Egypt. More than one third of apparently healthy liver donors in Egyptian study show steatosis in their routine preoperative biopsy (doi: 10.3978/j.issn.2304-3881.2015.10.03). Alternatively, ultrasonography is used for diagnosis of fatty liver. Fatty liver, as gross appearance of steatosis, is highly prevalent in Egypt to the level that about 16% Egyptian school children have fatty liver (doi.org/10.9734/ijtdh/2019/v36i130136). This characteristic finding in Egypt is due eating high fat diet in addition to other cultural habits. Steatosis is one of the characteristic triad to diagnose chronic HCV histologically (doi.org/10.1016/S1590-8658(11)60589-6). We added a statement to the discussion that while biopsies were not performed on healthy controls or CLD patients, from this body of literature we can estimate that all patients had likely steatosis but only around one third of the healthy controls might suffer from steatosis.
Li et al. (doi: 10.18632/oncotarget.23494) could not identify significant correlations between hepatic and serum lipids. They postulated “that hepatic and serum lipid signatures of HCC have to be considered as mostly independent, and the results imply potential roles of PEp species, particularly PEp (36:4) and (40:6), as serum biomarkers for HCC diagnosis and progression.” This paper should be discussed.
Response: We disscussed this paper as an example of scarce studies that investigate lipidomics in both blood and tissues in the same HCC patients. Specifically, the study PMID 29435160 by Lu et al (Oncotarget 2018) is indeed quite interesting, as it at least compared liver resections and blood, but studied a different patient cohort than ours. Only 1/50 patients in the Lu et al study was positive for HCV virus, while in our study, the majority of patients were positive for HCV when presenting to the clinic. The Lu et al study did not include other patients with CLD. In comparison to healthy controls, Lu et al found saturated blood triacylglycerides down-regulated, whereas we did not find such comparisons particularly interesting: the clinical question is the differentiation of HCC patients to CLD patients, not to healthy controls. Nevertheless, in our HCV-dominated cohort, saturated blood triacylglycerides were still higher in HCC than in healthy controls (Figure 4, 6). Similarly, in our HCV-dominated cohort, blood plasmalogens were increased compared to healthy controls (Lu et al: down), and free fatty acids were not even examined by Lu et al. In paired tumor/non-tumor tissues, Lu et al found increased triacylglycerides whereas in our HCV-dominated tumors, triacylglycerides were found decreased (but not significant). We found significant decreases in ceramides, Lu et al did not report such finding. One of the few similarities were significantly decreased phosphatidylglycerides (PG) which we also now added to the discussion section (lines 319-328).
A recent work suggested that the CE/free cholesterol ratio is specifically induced in HCC. Was this also found in the present cohort?
Response: While our data set did not include free cholesterol, but find this an important comment for future analyses that we will conduct.
Round 2
Reviewer 2 Report
authors have corrected the manuscript accordingly